# Association between functional dentition and ultra-processed food consumption in Brazilian adults: A cross-sectional study

Orlando Luiz do Amaral Junior[1], Maria Laura Braccini Fagundes[1],
Augusto Bacelo Bidinotto[2], Matheus Neves[2], Fernando Neves Hugo[3]*,
Jessye Melgarejo do Amaral Giordani[1]

1 Postgraduate Program in Dental Sciences, Federal University of Santa Maria, Santa Maria, Brazil,
2 Department of Preventive and Social Dentistry, Federal University of Rio Grande do Sul, Porto Alegre,
Brazil, 3 College of Dentistry, Department of Epidemiology and Health Promotion, New York University,
New York, New York, United States of America

* fnh9064@nyu.edu

journal.pone.0325838

University of Medical Sciences School of
Dentistry, ISLAMIC REPUBLIC OF IRAN

**Peer Review History:** PLOS recognizes the
benefits of transparency in the peer review
process; therefore, we enable the publication
of all of the content of peer review and
author responses alongside final, published
articles. The editorial history of this article is
available here: https://doi.org/10.1371/journal.
pone.0325838

## Abstract

### Objective

This study aimed to verify if there is an association between functional dentition and
ultra-processed diet in Brazilian adults.

### Methods

This cross-sectional study used data from the National Health Survey (PNS) 2019.
For the present study, we included only adults aged 18–49 (n = 50,146). The outcome
was the ultra-processed food consumption score, calculated based on the sum of
positive responses regarding the consumption of 10 subgroups of ultra-processed
foods on the previous day. These subgroups represent foods widely consumed in
Brazil, and the scoring method has been previously validated for this purpose. The
main predictor used was the variable 'functional dentition', categorized into individu-
als with less than 20 teeth and individuals with 20 or more teeth (functional dentition).
Multivariate analysis was performed using the Poisson regression model with robust
variance to estimate crude and adjusted prevalence ratios (PR).

### Results

It was observed that the presence of functional dentition (20 or more teeth) was not
associated with a greater consumption of an ultra-processed diet (RP = 0.97 [95% CI:
0.91–1.05]), $p$ = 0.578).

**Data availability statement:** The NHS data are available for public access and use at the official website of the Brazilian Institute of Geography and Statistics (IBGE) (https://www.ibge.gov.br/estatisticas/sociais/saude.html).

**Funding:** This work was supported by New York University, which provided financial assistance to cover the article processing charges (to FNH). In addition, OLAJ received a scholarship from the Brazilian Federal Agency for Support and Evaluation of Graduate Education (CAPES – Coordenação de Aperfeiçoamento de Pessoal de Nível Superior). These sources of support had no role in the study design, data collection, analysis, decision to publish, or preparation of the manuscript.

**Competing interests:** The authors have declared that no competing interests exist.

## Conclusion

The lack of a significant association between functional dentition and ultra-processed foods consumption suggests that social determinants might play a more important role in shaping dietary habits. The implementation of public policies that promote equitable access to healthy foods and preventive oral health care could contribute to overall health improvements and inequalities reduction.

## Introduction

Ultra-processed foods are industrial formulations, that seek to imitate the sensorial qualities of minimally processed foods and tend to be rich in oils, sugar, salt, and additives [1,2]. Diets with a high amount of ultra-processed foods tend to be nutritionally unbalanced, harmful to health, and highly associated with hypertension and obesity [3,4]. Individual and environmental aspects like socioeconomic position, education level, cultural and social pressures, apart from health status, can influence food choices [5,6]. Previous studies suggest an increase in the consumption of ultra-processed foods, influenced by their lower price and widespread supply [1].

Over the last half-century, public health nutrition, through epidemiological research, has identified potential dietary contributors capable of negatively influencing population health [7]. Regarding oral health conditions and food choices, it is known that people with severe tooth loss can report higher levels of chewing impairment and greater difficulty in eating food like fruits and vegetables [8]. Poor dental status can be adversely related to a poor diet, with those with severe tooth loss being at a nutritional disadvantage compared with dentate individuals, which implies a potential association between nutrient-rich foods and a higher number of present teeth [9]. Studies examining oral health variables like the number of teeth and overall diet quality, like consuming ultra-processed foods, using large adult population samples are still rare [5]. At the moment, no study has investigated the association between oral status (number of teeth) and the intake of ultra-processed foods in a nationwide sample of Brazilian adults. Although previous studies have demonstrated the health risks associated with ultra-processed foods, the literature commonly associates dietary patterns with masticatory capacity and functional dentition [7,9]. However, it remains unclear whether these factors are associated with the consumption of ultra-processed foods after considering social and economic determinants [10]. The assumption that the consumption of these foods is so "pervasive" that it does not relate with functional dentition underscores the importance of investigating health inequities.

Data regarding the importance of oral health to ensure proper eating behaviors and diet quality are fundamental to the formulation of strategies to reduce chronic disease burden. Better oral health may improve overall diet quality, which in its hand contributes to better health outcomes [5,8]. Therefore, this study aimed to verify if there is an association between functional dentition, as represented by having 20 or more natural teeth, and ultra-processed food consumption in Brazilian adults, using

the data from the National Health Survey (PNS) 2019. The null hypothesis is that functional dentition is not associated with lower consumption of ultra-processed food. Alternatively, we hypothesize that functional dentition is significantly associated with lower consumption of ultra-processed foods.

## Methodology

### Design and scenario

This cross-sectional research utilized data from the PNS 2019, a population-based survey that represents the Brazilian population living in private households. The survey allows for estimating data across urban and rural areas, major national regions, all federation units (UFs), capitals, and metropolitan regions. A three-stage cluster sampling method was employed. The primary sampling units (PSUs) were census tracts or groups of tracts, the second stage involved private households chosen through simple random sampling, and the third stage included residents aged 15 years or older [11].

To calculate the sample size with the required precision for the estimates, indicators from the 2013 PNS edition were considered, including data on chronic conditions, violence, healthcare service use, health insurance, smoking, physical activity, and alcohol consumption, among others. Data collection took place between August 2019 and March 2020, using mobile devices. When arriving at the selected households, interviewers explained the objectives, data collection procedures, and the importance of participation in the survey. A list of all individuals residing in the household was prepared, after which one resident aged 15 years or older was randomly selected to respond to the individual questionnaire. Interviews were conducted at a time convenient for the residents, with two or more visits planned per household. In total, 94,114 interviews were completed, with a non-response rate of 6.4%. Additional methodological details can be found elsewhere [11]. For this study, only adults aged 18–49 years were included in the analysis. This age range was selected to minimize the potential for survivorship bias in the study, as individuals aged 50 years and above who remain in the sample often represent a subgroup with better overall health, including oral health, compared to their peers. This selective survival can distort findings by underestimating the true prevalence of oral health issues and their association with dietary habits.

### Ethical aspects

The PNS data are available for public access and use at the official website of the Brazilian Institute of Geography and Statistics (IBGE) (https://www.ibge.gov.br/estatisticas/sociais/saude.html). The PNS was approved by the National Research Ethics Commission (CONEP 3.529.376). An Informed Consent Form was obtained from all participants at the time of the interview [11].

### Participants and sample selection

The PNS 2019 interviewed 88,531 individuals aged 15 years or older. The participants, who were randomly selected, answered an individual questionnaire. For this study, we included only adults aged 18–49 years (n = 50,146) [11]. The research sample excluded households located in special or sparsely populated census sectors, such as Indigenous groups, barracks, military bases, accommodation, camps, boats, penitentiaries, penal colonies, prisons, jails, long-term institutions for older adults, networks of integrated care for children and adolescents, convents, hospitals, agricultural villages for settlement projects, and quilombola groups [11]. This exclusion was necessary due to the unique characteristics, accessibility challenges, and specific living conditions of these populations, which require tailored methodologies to ensure accurate and representative data collection. The data collection was coordinated by the IBGE and carried out by field researchers, supervisors, and state coordinators. Data was collected face-to-face using mobile devices at the participants' homes. More details about the PNS methodology are published elsewhere [11].

## Variables

**Outcome.**  The outcome of this study was the ultra-processed food consumption score, calculated based on the sum of positive responses regarding the consumption of 10 subgroups of ultra-processed foods on the previous day. These subgroups represent foods widely consumed in Brazil, and the scoring method has been previously validated for this purpose. In the PNS 2019, participants responded with "yes" or "no" to the following questions: "Yesterday, did you drink or eat: (1) Soft drink?; (2) Fruit juice drink in a can or box or prepared from a powdered mix?; (3) Chocolate powder drink or flavored yogurt?; (4) Packaged salty snacks or crackers?; (5) Sandwich cookies or sweet biscuits or packaged cake?; (6) Ice cream, chocolate, gelatin, flan, or other industrialized dessert?; (7) Sausage, mortadella or ham?; (8) Loaf, hot dog or hamburger bun?; (9) Margarine, mayonnaise, ketchup or other industrialized sauces?; (10) Instant noodles, instant powdered soup, frozen lasagna or other frozen ready-to-eat meal? The score of ultra-processed food consumption of each participant was calculated by adding up the positive answers given to these questions regarding consumption on the day before the interview, which can vary from 0 to 10 points. Subsequently, the variable was dichotomized at a cutoff point of <4 (individuals who have a low consumption of ultra-processed foods) and ≥5 (individuals who have a greater consumption of ultra-processed foods). The choice for this categorization was based on a previous study [12].

**Covariates.**  A Directed Acyclic Diagram (DAG) was made (Fig 1) to identify possible confounders in the relationship between exposure and outcome [13]. The following covariates were included: zone of residency (urban/rural); sex (male and female); skin color (white/non-white); age (18–29/30–39/40–49); per capita household income (less than 1 minimum wage (SM)/ 1–3 MW/ 3–5 MW and more than 5 MW); Education (0–8 years of study/ 9 or more years of study). As a behavioral variable, the use of dental services was used, considering the last 12 months, categorized as (did not use the service in the last 12 months/ used the service in the last 12 months). The main predictor used was the self-reported variable 'functional dentition', categorized into individuals with fewer than 20 teeth and individuals with 20 or more teeth (functional dentition). This categorization, which has been widely used in population-based surveys due to its practicality and relevance for assessing oral health, was adopted from a previous study [14].

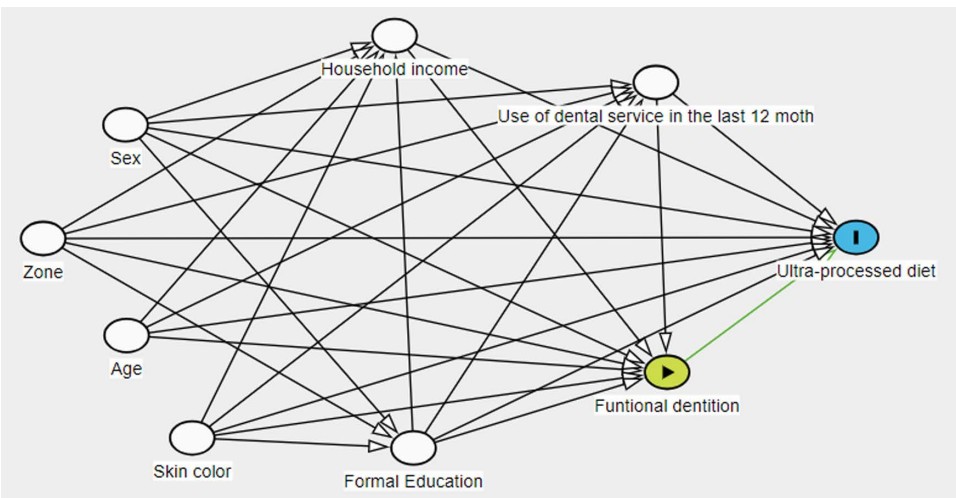

**Fig 1.  Directed acyclic graph (DAG) illustrating hypothesized confounders in the relationship between functional dentition and ultra-processed food consumption.**

## Statistical analysis

Data analysis adopted sampling weights for primary sampling units, households, and selected residents, following complex survey sampling [12,22]. The analyses were carried out in Stata software, version 14.0 (College Station, TX), using the survey module.

First of all, a descriptive analysis was performed to verify the general prevalence of sample characteristics and proportions by consumption of ultra-processed foods. Subsequently, multivariate analysis was performed using the Poisson regression model and robust variance to estimate crude and adjusted prevalence ratios (PR), their respective 95% confidence intervals (95% CI), and a significance level of 5%.

## Results

In total, 50,146 individuals were evaluated, of whom 15.7% consumed more ultra-processed foods. Table 1 describes the frequencies, proportions and unadjusted prevalence ratios of ultra-processed diet consumption according to

**Table 1. Characteristics of the sample and high consumption of ultra-processed food by socioeconomic factors. National Health Survey (PNS) 2019, Brazilian adults aged 18-49 years (n = 50,146).**

| Sample Characteristics | Weighted (%)* | The proportion of ultra-processed diet per exposure (CI 95%)[a] | Unadjusted PR[a] (95% CI)* |
|---|---|---|---|
| *Socioeconomic factors* | | | |
| Zone | | | |
| Rural | 13.8 | 19.4 (18.7 - 20.2) | **1** |
| Urban | 86.2 | 9.9 (9.0–10.9) | **1.08 (1.07–1.09)**[b] |
| Sex | | | |
| Male | 47.8 | 19.7 (18.7–20.7) | **1** |
| Female | 52.1 | 16.6 (15.8–17.5) | **0.97 (0.96 – 0.98)**[b] |
| Skin Color | | | |
| White | 39.8 | 19.4 (18.3–20.5) | **1** |
| Not White | 60.1 | 17.3 (16.5–18.0) | **0.98 (0.97–0.99)**[b] |
| Age groups (in years) | | | |
| 18-29 | 36.7 | 23.0 (21.7–24.3) | **1** |
| 30-39 | 34.2 | 17.7 (16.7–18.7) | **0.95 (0.94–0.97)**[b] |
| 40-49 | 29.6 | 12.7 (11.8–13.8) | **0.91 (0.90–0.92)**[b] |
| Formal education | | | |
| ≤ 8 Years | 29.4 | 14.3 (13.3–15.3) | **1** |
| > 8 Years | 70.5 | 19.7 (18.9–20.5) | **1.04 (1.03–1.05)**[b] |
| Household income[a] | | | |
| Up to 1 Minimum wage (MW) | 56.1 | 15.7 (14.4–17.0) | **1** |
| More than 1 up to 3 (MW) | 34.1 | 20.5 (18.7–22.4) | **1.02 (1.01–1.04)**[b] |
| More than 3 up to 5 (MW) | 5.4 | 18.3 (15.7–21.2) | 1.01 (0.98–1.03) |
| More than 5 (MW) | 4.1 | 13.1 (10.7–16.0) | **0.96 (0.94–0.99)**[b] |
| Use of dental service in the last 12 months | | | |
| No use | 45.3 | 17.4 (16.5–18.4) | **1** |
| Use | 54.6 | 18.7 (17.8–19.6) | **1.01 (1.00–1.02)**[b] |
| Functional dentition | | | |
| 19 or less teeth | 1.5 | 13.8 (7.4–24.2) | 1 |
| 20 or more teeth | 98.4 | 18.2 (17.5–18.8) | 0.96 (0.89–1.03) |

*Taking into account the sample weight.

[a]95% CI, 95% confidence interval.

[b]*p*-value < 0.05

socioeconomic factors and demographic characteristics. Greater ultra-processed food consumption was more prevalent among urban residents (19.4%) compared to rural residents (9.9%). Males (19.7%) showed a greater prevalence of the outcome than females (16.6%). White individuals (19.4%) had a slightly higher prevalence than those of other skin colors (17.3%). High ultra-processed food consumption was greater among young adults aged 18–29 years (23.0%) and progressively decreased with age, being lowest among those aged 40–49 years (12.7%).

Individuals with more than 8 years of formal education had a higher prevalence of ultra-processed foods consumption (19.7%) compared to those with 8 years or less of schooling (14.3%). Prevalence was also higher among those with household incomes between 1 and 3 minimum wages (20.5%) and decreases as income increases, being lowest among those earning more than 5 minimum wages (13.1%). Regarding dental service use, the difference was minimal, with (17.4%) among those who did not use services and (18.7%) among those who did in the last 12 months. Individuals with functional dentition (20 or more teeth) showed a higher prevalence of ultra-processed diet consumption (18.2%) compared to those with 19 or fewer teeth (13.8%), but this difference was not statistically significant.

Regarding the proposed association (Table 2), it was possible to observe that having 20 or more natural teeth (functional dentition) was not associated with higher ultra-processed consumption (PR = 1.02, 95% CI: 0.94–1.09, p = 0.578).

## Discussion

This study aimed to investigate the association between functional dentition and the consumption of ultra-processed foods in a representative sample of Brazilian adults, using data from the PNS 2019. Although previous studies have suggested an association between tooth loss and diet [15,16], our results did not find a significant relationship between the presence of 20 or more teeth and lower consumption of ultra-processed foods. Thus, we reject the pre-specified hypothesis that functional dentition is significantly associated with lower ultra-processed food consumption in this population. These findings suggest that functional dentition alone may not be a determining factor in the consumption of such foods, raising important questions about the complexity of this relationship. It should be considered that even though for adolescents functional dentition does not appear to be associated with diet, for Brazilian older adults the use of dentures was associated with low vegetable consumption and a higher percentage of body fat [17–19].

Studies have frequently associated oral health, especially having an adequate number of teeth, with greater consumption of healthy foods, such as fruits and vegetables, which require more masticatory effort [15,20]. However, ultra-processed foods are generally easy to chew and quick to consume, regardless of oral status.3 Evidence shows that compromised dentition can negatively affect diet quality [21,22]. Nonetheless, our findings indicate that tooth loss has limited influence on the consumption of ultra-processed foods, likely due to their soft texture and ease of ingestion. This suggests that the intake of such products is less affected by masticatory limitations in this population. These results

**Table 2. Unadjusted and adjusted association between ultra-processed diet and functional dentition in Brazilian adults aged 18-49 years (n = 50,146) through Poisson regression with robust variance.**

| Functional dentition | Ultra-processed diet | |
|---|---|---|
| | Unadjusted PRᵃ (95% CIᵇ)* | Adjusted‡ PRᵃ (95% CIᵇ)* |
| Functional dentition | | |
| 19 or less teeth | 1 | 1 |
| 20 or more teeth | 1.03 (0.96–1.11) | 1.02 (0.94–1.09) |

ᵃPR, prevalence ratio;

ᵇ95% CI, 95% confidence interval.

*Taking into account the sample weight.

‡Adjusted for: zone, sex, skin color, age, formal education, household income, and use of dental service in the last 12 months.

underscore the need for public health strategies that improve dietary quality by reducing the availability and attractiveness of ultra-processed foods, particularly among socioeconomically vulnerable groups.

It is important to consider that socioeconomic and behavioral factors may play a more significant role in food choice than the number of teeth [20]. The inclusion of covariates such as income, education, urban/rural setting, and use of dental services in the analysis may have diluted the association between functional dentition and diet, as these factors can act as direct determinants in the choice of ultra-processed foods [7,23]. The results also revealed that lower levels of education and income were consistently associated with higher consumption of ultra-processed foods, even after adjusting for dental status and other covariates, reflecting a clear social gradient. This finding reinforces the notion that structural determinants of health strongly influence dietary behavior, independent of oral health status (Appendix I). Eating behavior, shaped by cultural norms and economic constraints, may mediate this relationship, suggesting that a more comprehensive exploration of these pathways would be valuable in future research [24–26].

As people experience greater tooth loss, they may adapt their diets by choosing foods that are easier to chew, such as ultra-processed foods, which demand minimal masticatory effort. As a result, dentition becomes less relevant for the consumption of these foods, as ultra-processed foods are more suitable for reduced masticatory conditions. On the other hand, dentition may play a more significant role in diets rich in minimally processed foods, such as fruits, vegetables, and meats, whose physical characteristics require greater chewing capacity [23]. Furthermore, it is important to consider that dentition may have a greater impact on overall diet quality than on the specific intake of ultra-processed foods.

The relationship between functional dentition and the ultra-processed foods consumption can be understood through the lens of the Common Risk Factor Approach, which recognizes that multiple health problems share underlying determinants, such as socioeconomic and behavioral factors [27]. Although our results did not identify a significant direct association between the number of teeth and the outcome, it was observed that characteristics such as education level, income, and place of residence influence both oral health and dietary patterns [28]. These common risk factors, such as unfavorable socioeconomic conditions, affect access to healthy foods and quality oral care [27,29]. Therefore, oral health and diet appear to be interconnected by external factors, such as social inequalities [30,31].

This study has several limitations that should be considered when interpreting the findings. First, as a cross-sectional study, it is not possible to establish a causal relationship between functional dentition and the consumption of ultra-processed foods. Additionally, dietary intake was measured through self-report, which may introduce recall bias, as participants may not accurately remember the foods consumed the day before the interview. Another limitation is that the study focused on an adult sample aged 18 to 49 years, which may limit the generalizability of the findings to other age groups, particularly the older adults, who tend to have more oral health problems. Finally, the dichotomous categorization of the ultra-processed food consumption variable may not capture the nuances of dietary patterns, and future studies could benefit from a more detailed analysis of the quantities and types of ultra-processed foods consumed. To address potential sources of bias, the study adjusted for socioeconomic, demographic, and behavioral factors to minimize confounding effects. However, residual confounding remains a possibility due to the cross-sectional nature of the study and the reliance on self-reported dietary data. On a positive note, the study offers valuable insights into the dietary behaviors of a crucial segment of the adult population in Brazil. Despite its limitations, the large and diverse sample provides a robust foundation for understanding the relationship between diet and oral health. These findings can inform targeted public health interventions and underscore the need for comprehensive dietary assessments in future research.

## Conclusion

In summary, this study underscores the relevance of addressing common risk factors when examining the relationship between oral health and dietary patterns. The absence of an association between functional dentition and the consumption of ultra-processed foods suggests that social determinants may exert a stronger influence on dietary behaviors than

oral conditions. Public policies that ensure equitable access to healthy foods and preventive oral health services are essential for improving diet quality and reducing health inequalities.

## Appendix I

Adjusted association between ultra-processed diet and functional dentition in Brazilian adults aged 18-49 years (n=50,146) through Poisson regression with robust variance.

| Sample Characteristics | Adjusted PR[a] (95% CI)* |
|---|---|
| **Socioeconomic factors** | |
| Zone | |
| Urban | 1 |
| Rural | 1.07 (1.06–1.08)[b] |
| Sex | |
| Male | 1 |
| Female | 0.96 (0.95 – 0.97)[b] |
| Skin Color | |
| White | 1 |
| No White | 0.98 (0.97–0.99)[b] |
| Age (years) | |
| 18-29 | 1 |
| 30-39 | 0.95 (0.94–0.97)[b] |
| 40-49 | 0.92 (0.90–0.93)[b] |
| Formal education | |
| ≤ 8 Years | 1 |
| > 8 Years | 1.01 (1.00–1.03)[b] |
| Household income [a] | |
| Up to 1 Minimum wage (MW) | 1 |
| More than 1 up to 3 (MW) | 1.01 (0.99–1.02) |
| More than 3 up to 5 (MW) | 0.98 (0.96–1.01) |
| More than 5 (MW) | 0.94 (0.92–0.96)[b] |
| Use of dental service in the last 12 months | |
| No use | 1 |
| Use | 1.00 (0.99–1.02) |
| Functional dentition | |
| 19 or less teeth | 1 |
| 20 or more teeth | 1.02 (0.94–1.09) |

[a] PR, prevalence ratio; [b]95% CI, 95% confidence interval.
* Taking into account the sample weight.
[b] p-value < 0.05

## Author contributions

**Conceptualization:** Orlando Luiz do Amaral Júnior, Maria Laura Braccini Fagundes, Matheus Neves, Fernando Neves Hugo, Jessye Melgarejo do Amaral Giordani.

**Data curation:** Orlando Luiz do Amaral Júnior, Maria Laura Braccini Fagundes.

**Formal analysis:** Orlando Luiz do Amaral Júnior.

**Investigation:** Orlando Luiz do Amaral Júnior, Maria Laura Braccini Fagundes.

**Methodology:** Orlando Luiz do Amaral Júnior, Augusto Bacelo Bidinotto, Fernando Neves Hugo.

**Supervision:** Fernando Neves Hugo, Jessye Melgarejo do Amaral Giordani.

**Validation:** Orlando Luiz do Amaral Júnior, Fernando Neves Hugo.

**Visualization:** Maria Laura Braccini Fagundes, Matheus Neves, Fernando Neves Hugo, Jessye Melgarejo do Amaral Giordani.

**Writing – original draft:** Orlando Luiz do Amaral Júnior, Maria Laura Braccini Fagundes, Augusto Bacelo Bidinotto, Matheus Neves, Fernando Neves Hugo, Jessye Melgarejo do Amaral Giordani.

**Writing – review & editing:** Orlando Luiz do Amaral Júnior, Maria Laura Braccini Fagundes, Augusto Bacelo Bidinotto, Matheus Neves, Fernando Neves Hugo, Jessye Melgarejo do Amaral Giordani.

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
