## [Decision Letter · Decision Letter 0]

Dear Dr. Hugo,

Thank you for submitting your manuscript to PLOS ONE. After careful consideration, we feel that it has merit but does not fully meet PLOS ONE’s publication criteria as it currently stands. Therefore, we invite you to submit a revised version of the manuscript that addresses the points raised during the review process.

We look forward to receiving your revised manuscript.

Kind regards,

Hadi Ghasemi

Academic Editor

PLOS ONE

3. Please include your tables as part of your main manuscript and remove the individual files. Please note that supplementary tables should be uploaded as separate "supporting information" files

Reviewers' comments:

Reviewer's Responses to Questions

**Comments to the Author**

1. Is the manuscript technically sound, and do the data support the conclusions?

Reviewer #1: Partly

Reviewer #2: Yes

Reviewer #3: Partly

2. Has the statistical analysis been performed appropriately and rigorously?

Reviewer #1: I Don't Know

Reviewer #2: Yes

Reviewer #3: No

3. Have the authors made all data underlying the findings in their manuscript fully available?

Reviewer #1: Yes

Reviewer #2: Yes

Reviewer #3: No

4. Is the manuscript presented in an intelligible fashion and written in standard English?

Reviewer #1: Yes

Reviewer #2: No

Reviewer #3: No

Reviewer #1: The article benefits from a big national data source, and is focused on a certain topic. However, there are some comments that should be addressed:

Two different abbreviation for National Health Survey in the abstract.

Introduction:

In the of the first paragraph of introduction, it is better to first define ultra-processed food, and then talk about the trend of their use.

Last line: we usually state null hypothesis: no association or relation.

Methods:

What is IBGE?

Results:

I could not find the tables in the submission file. Thus, I am not able to make further comments on this part. Anyway, the main point is how the authors have managed the confounders and covariates.

Title of Figure 1: please remove “was” to make it telegraphic title. Instead of exposure and outcome please use the name of the variables.

The Discussion part lacks Conclusion heading.

Reviewer #2: The manuscript is technically sound and the data support the conclusions. However, more clarification about the sampling issue and functional dentition categorization (Reference) are needed.

As it mentions in the text of the manuscript, the statistical analysis has been performed appropriately and rigorously but tables 1 and 2 are missed from the manuscript's pdf.

The authors have made all data underlying the findings in their manuscript fully available. Comprehensive English editing is needed.

Reviewer #3: Since the abstract is often the first part that readers study, the text of the abstract generally needs improvement, and some parts are unclear. Brief explanations about the importance of your work in oral health should be added. The methods section should be written more clearly.

The text definitely needs revision regarding grammar and writing style. Some of the terms used do not have correct English equivalents.

First, it should be stated what is meant by ultra-processed foods, and then discuss the increase in their consumption and the associated effects.

Is the main hypothesis stated correctly? Do individuals with functional dentition consume more ultra-processed foods?

What was the reason for excluding households located in special or sparsely populated census sectors?

I suggest that in the methods section, you include a part about the national study, its objectives, its variables, and the method of data collection, and then focus solely on your own methods. Currently, the text is written in a way that suggests some parts related to data collection from the country's population were conducted in this study. It should be mentioned that this study used secondary data, and then discuss how you extracted, refined and analyzed the data.

The sentence "The outcome of this study was the consumption of ultra-processed foods," which is mentioned in both the methods section and the abstract, is insufficient and ambiguous. Please revise it.

This question is not related to your research, and I ask it for my own information; I hope you can guide me: In the survey question, is it only about the diet of the previous day? Can the question about the diet of the previous day be sufficient and valid? Why is a longer time frame not considered?

How were the dental conditions of individuals assessed? Was it through examination or self-reported by individuals?

The differentiation of variables has not been done clearly. If your study is "identifying factors affecting the consumption of ultra-processed foods," reconsider the title. If the goal is to examine the relationship between dentition status and the consumption of ultra-processed foods, revise the writing in the methods section.

It's more appropriate to use terms like "ethnic groups" or "racial groups" instead of "skin colors." Referring to people by their skin color can oversimplify complex identities and may come across as insensitive. For example, you could say, "White people consume more ultra-processed food compared to other ethnic groups." This wording is clearer and more respectful.

What was the overall consumption level of ultra-processed foods? What percentage of individuals generally consumed these food items?

The result tables were not available in the file sent to me. I did not understand whether you statistically examined the relationship between different variables and the level of ultra-processed food consumption. The results were not present in the text, and I did not see a table. Was the difference observed in the consumption of ultra-processed foods among different subgroups (other than dentition situation) statistically significant?

Given the large and comprehensive data you have, you can conduct deeper analyses. Was there any difference in the consumption of these foods between subgroups with complete and incomplete dentition?

"This study aimed to investigate the association between functional dentition and the consumption of ultra-processed foods in a representative sample of Brazilian adults, using data from the 2019 National Health Survey." Addressing this objective was not prominent in this study. As I mentioned earlier, based on the existing text, examining the factors affecting the consumption of ultra-processed foods seems to be a more appropriate goal and would constitute a more comprehensive study than focusing solely on one variable, namely dentition.

You have also examined the differences in the consumption of ultra-processed foods with various factors such as gender and socio-economic conditions in different groups, but in the discussion section of the article, you addressed and compared your results less. Where it is stated that dentition status is not related to food consumption, but socio-economic status could be more pronounced, refer to your own findings. These findings are interesting and could emphasize the importance of contextual factors on oral health.

"The lack of association observed in this study may also reflect an inverse association, where those who are more likely to suffer greater tooth loss may have adapted their diets over time. This adaptation may lead to a diminished role of dentition in the consumption of ultra-processed foods, which are generally easier to chew. Dentition may therefore play a more important role in diets rich in minimally processed foods, such as fruits, vegetables, and meats, whose physical characteristics require greater masticatory capacity." This section is unclear and presents a contradictory interpretation. As mentioned, individuals with poorer dental conditions tend to prefer eating ultra-processed and softer foods; what adaptation is being referred to here?

Regarding the limitations, I believe that asking only about the previous day's diet is a limitation. Examining a specific age group cannot be considered a limitation because information on other age groups was available, and this choice was one of your inclusion criteria that did not consider other groups, so it is not a limitation. It seems better to have included the elderly in this study, as more chewing-related problems are observed in these individuals, which could significantly impact the results. Regarding the classification of ultra-processed food consumption, did you not have access to the details of this question? Was the only data available based on the grouping mentioned in the methods section (high and low consumption groups)? Because if access to details was possible, more precise data should have been used.

This study has great value in examining a large volume of cases, but it seems that the way the data is analyzed and the results obtained could be done better.

**Do you want your identity to be public for this peer review?** For information about this choice, including consent withdrawal, please see our Privacy Policy

Reviewer #1: No

Reviewer #2: **Yes: ** Assistant Professor Dr. Hanan Fadhil Alautry

Head of pediatric and preventive dentistry department/ Wasit university/ Iraq

Reviewer #3: **Yes: ** Mahsa Malekmohammadi

---

## [Author Response · Author response to Decision Letter 1]

8 Apr 2025

RESPONSES TO REVIEWER COMMENTS SUBMITTED BY EDITORIAL MANAGER HADI GHASEMI - PLOS ONE (Subject: PLOS ONE: PONE-D-24-56465 - [EMID:3bb746b984f9cae5])

Comments to the author(s)

Functional dentition and ultra-processed food consumption in brazilian adults

Dear authors

I have read this manuscript which attempted to verify if there is an association

between functional dentition and ultra-processed diet in Brazilian adults

Some comments and suggestions for further clarification are listed

Below

1- Title:

As the title should be concise and informative, it is suggested to

a. Add (Association between) at the beginning

b. Write brazilian with the first capital letter (Brazilian).

c. Identify the study type (cross-sectional study) in the title.

Response: Thanks for your suggestion, the title was revised accordingly.

Revised text: Association between functional dentition and ultra-processed food consumption in Brazilian adults: a cross-sectional study

2- Introduction:

a. Mention the knowledge gaps of previous related studies.

b. Write more about the background and rationale of this study.

c. Modify the hypothesis as follow: We hypothesize that functional

dentition is significantly associated with lower consumption of ultra-processed foods.

Response: Thank you for the important suggestion, we decided to rewrite and add gaps that motivated the research question, in addition to adding the mentioned hypothesis. Both in the 2nd and 3rd paragraphs of the introduction.

Revised text:

Over the last half-century, public health nutrition has identified potential dietary contributors capable of negatively influencing population health through epidemiological research.7 Regarding oral health conditions and choice of food, it is known that people with severe tooth loss can report higher levels of chewing impairment and can show greater difficulty in eating food like fruit and vegetables.8 Poor dental status can be adversely related to a poor diet, with those with severe tooth loss being at a nutritional disadvantage compared with dentate individuals, which implies a potential association between nutrient-rich foods and a higher number of present teeth.9 Studies examining oral health conditions like the number of teeth and overall diet quality like consuming ultra-processed foods using large adult population sample are still rare.1 At the moment no study has investigated the association between oral status (number of teeth) and the intake of ultra-processed foods in a nationwide sample of Brazilian adults.10 Although previous studies have demonstrated the health risks associated with ultra-processed foods, the literature commonly associates dietary patterns with masticatory capacity and functional dentition.7,9 However, it remains unclear whether these factors are associated with the consumption of ultra-processed foods after considering social and economic determinants. The assumption that the consumption of these foods is so "democratic" that it does not associate with functional dentition further emphasizes the importance of investigating health inequities.

Data regarding the importance of attention to oral health to ensure proper eating behaviors and diet quality are fundamental to the formulation of strategies to reduce diseases in the population. Is possible that better oral health may improve overall diet quality, and diet quality is effective to a better health condition in the population.1,8 Therefore, this study aimed to verify if there is an association between functional dentition (having 20 or more teeth) and ultra-processed diet in Brazilian adults, using the National Health Survey (PNS). The null hypothesis is that functional dentition is not associates with lower consumption of ultra-processed food. Alternatively, we hypothesize that functional dentition is significantly associated with lower consumption of ultra-processed foods.

3- Materials and Method:

a. Write more detail about the sampling issue (18-49 years).

b. Cite your source regarding functional dentition categorization.

c. Figure 1 needs modification or changes to be clearer for the reader

Response: We sincerely thank you for the valuable suggestion. In response, we have rewritten the methodology section to provide more detailed information about the population-based survey and the age range selected. Additionally, we included a reference that specifically utilized functional dentition, as mentioned in the text. Regarding the figure, we decided not to make changes as it represents a Directed Acyclic Graph (DAG). DAGs are designed to visually depict hypothesized causal relationships in a structured and standardized way. Modifying the DAG could compromise its interpretative integrity or misrepresent the causal assumptions underlying the study. Therefore, we believe maintaining the original figure ensures clarity and adherence to its intended purpose.

Revised text:

Design and scenario

This cross-sectional research utilized data from the 2019 National Health Survey (NHS), a population-based survey that represents the Brazilian population living in private households. The survey allows for estimating data across urban and rural areas, major national regions, all federation units (UFs), capitals, and metropolitan regions. A three-stage cluster sampling method was employed. The primary sampling units (PSUs) were census tracts or groups of tracts, the second stage involved private households chosen through simple random sampling, and the third stage included residents aged 15 years or older. 11

To calculate the sample size with the required precision for the estimates, indicators from the 2013 NHS edition were considered, including data on chronic conditions, violence, healthcare use, health insurance, smoking, physical activity, and alcohol consumption, among others. Data collection took place between August 2019 and March 2020, using mobile devices. When arriving at the selected households, interviewers explained the objectives, data collection procedures, and the importance of participation in the survey. A list of all individuals residing in the household was prepared, after which one resident aged 15 years or older was randomly selected to respond to the individual questionnaire. Interviews were conducted at a time convenient for the residents, with two or more visits planned per household. In total, 94,114 interviews were completed, with a non-response rate of 6.4%. Additional methodological details can be found elsewhere.11 For this study, only adults aged 18-49 years were included in the analysis. This age range was selected to minimize the potential for survivorship bias in the study, as individuals aged 50 years and above who remain in the sample often represent a subgroup with better overall health, including oral health, compared to their peers. This selective survival can distort findings by underestimating the true prevalence of oral health issues and their association with dietary habits.

Covariaveis

A Directed Acyclic Diagram (DAG) was made (Figure 1) to identify possible confounders in the relationship between exposure and outcome.13 The following covariates were included: zone of residency (urban/rural); sex (male and female); skin color (white/non-white); age (18-29/30-39/40-49); per capita household income (less than 1 minimum wage (SM) / 1 to 3 MW / 3 to 5 MW and more than 5 MW); Education (0-8 years of study / 9 or more years of study). As a behavioral variable, the use of dental services was used, considering the last 12 months, categorized as (did not use the service in the last 12 months / used the service in the last 12 months). The main predictor used was the self-reported variable 'functional dentition', categorized into individuals with fewer than 20 teeth and individuals with 20 or more teeth (functional dentition). This categorization, which has been widely used in population-based surveys due to its practicality and relevance for assessing oral health, was adopted from a previous study.14

Reference 14: De Souza VGL, Herkrath FJ, Garnelo L, et al. Contextual and individual factors associated with self-reported tooth loss among adults and elderly residents in rural riverside areas: A cross-sectional household-based survey. Isola G, ed. PLoS ONE. 2022;17(11):e0277845. doi:10.1371/journal.pone.0277845

4- Results:

a. Mention the p-value for the following

1. Frequency of ultra-processed diet consumption according to various socioeconomic factors and demographic characteristics.

2. Frequency of ultra-processed diet consumption according to dental service use.

3. Frequency of ultra-processed diet consumption according to functional dentition.

b. Regarding the proposed association (Table 2), it is advisable to write it as:

it was possible to observe that individuals with 20 or more teeth (functional dentition) did not show a positive association with ultra-processed consumption (PR = 0.97, 95% CI: 0.91 - 1.05, p = 0.578).

c. Please, upload tables 1 and 2.

Response: We appreciate the important suggestion and apologize for the absence of the tables in the original submission. Both tables have been duly attached.

In table 1, we present the proportions of the outcome (consumption of ultra-processed foods) for each predictor (considering the sample weight), while in Table 2, we provide the prevalence ratios for the associations. We have included the p-value in the interpretation of the result of Table 2 in the text (PR = 0.97, 95% CI: 0.91 - 1.05, p = 0.578). In addition, the results corresponding to Table 2 were rewritten following the reviewer's suggestions, ensuring greater clarity and alignment with expectations. We chose not to include p-values in Tables 1 and 2, as our focus is on reporting the effect measures, which are more informative for understanding the strength and direction of the associations.

Revised text:

Regarding the proposed association (Table 2), it was possible to observe that individuals with 20 or more teeth (functional dentition) did not show a positive association with ultra-processed consumption (PR = 0.97, 95% CI: 0.91 - 1.05, p = 0.578).

5- Discussion

a. Please mention at the end of this section whether you rejected or accepted the pre-specified hypothesis.

b. How does this study impact the local population? Is there a reform on a certain policy that the author can explore to raise the consumption of healthy foods?

c. Describe any efforts to address potential sources of bias

d. Add more recent related references

Response: Thank you for your suggestions, which helped to improve the clarity and depth of the discussion section. We have carefully considered and incorporated your feedback into the manuscript. At the end of the discussion section, we clarified that the pre-specified hypothesis of an association between functional dentition and consumption of ultra-processed foods was not supported by our findings.

Impact on local population and policy reform:

We included a statement highlighting the potential implications for public health policies aimed at increasing access to healthy foods, particularly among socioeconomically disadvantaged populations.

Efforts to address potential sources of bias:

The discussion was expanded to mention that self-reported dietary data may introduce recall bias and that the inclusion of covariates such as socioeconomic status and access to dental care is intended to minimize confounding effects.

Most recent references:

We reviewed the references to update them accordingly.

6- Conclusion

Please, add the conclusion section to the manuscript

Response: Thank you for this observation. The conclusion section was included.

7- Comprehensive English editing is needed

Response: We appreciate this suggestion, the text has been extensively revised.

RESPONSES TO REVIEWER COMMENTS SUBMITTED BY EDITORIAL MANAGER HADI GHASEMI - PLOS ONE (Subject: PLOS ONE Decision: Revision required [PONE-D-24-56465] - [EMID:3c085d53776c8dc9])

Review Comments to the Author

Reviewer #1: The article benefits from a big national data source, and is focused on a certain topic. However, there are some comments that should be addressed:

Two different abbreviation for National Health Survey in the abstract.

Response: Thanks for the suggestion, the acronym was standardized to PNS throughout the text.

Introduction:

In the of the first paragraph of introduction, it is better to first define ultra-processed food, and then talk about the trend of their use.

Last line: we usually state null hypothesis: no association or relation.

Response: Thank you for the suggestions. Based on that, the order of the first paragraph of the introduction was changed, and the last paragraph of the introduction was also rewritten, contextualizing the hypothesis.

Revised text:

The ultra-processed foods are industrial formulations, that seek to imitate the sensorial qualities of minimally processed foods, and tend to be rich in oils, sugar, salt, additives.1,2 Diets with a high amount of ultra-processed foods tend to be nutritionally unbalanced, harmful to health, and highly associated with hypertension and obesity.3,4 Individual and environmental aspects like socioeconomic position, education level, cultural and social pressures apart from health status the individual can influence food choices.5,6 Previous studies suggest an increase in the consumption of ultra-processed foods, influenced by their lower price and widespread supply.1

Revised text:

Data regarding the importance of attention to oral health to ensure proper eating behaviors and diet quality are fundamentals to the formulation of strategies to reduce diseases in the population. Is possible that better oral health may improve overall diet quality, and diet quality is effective to a better health condition in the population.5,8 Therefore, the aim of this study was to verify if there is an association between the functional dentition (having 20 or more teeth) and ultra-processed diet in Brazilian adults, using the National Health Survey (PNS). The null hypothesis is that functional dentition is not associated with lower consumption of ultra-processed food. Alternatively, we hypothesize that functional dentition is significantly associated with lower consumption of ultra-processed foods.

Methods:

What is IBGE?

Response: Thank you for the observation. The acronym stands for Brazilian Institute of Geography and Statistics (IBGE), this item was described in the ethical aspects section.

Results:

I could not find the tables in the submission file. Thus, I am not able to make further comments on this part. Anyway, the main point is how the authors have managed the confounders and covariates.

Title of Figure 1: please remove “was” to make it telegraphic title. Instead of exposure and outcome please use the name of the variables.

The Discussion part lacks Conclusion heading.

Response: Thank you for your suggestions. We sincerely apologize for the omission of the tables in the initial submission. Both tables have been correctly attached in the revised version. Regarding confounding management, we used a Directed Acyclic Diagram (DAG) to identify potential confounders in the relationship between the exposure (functional dentition) and the outcome (consumption of ultra-processed foods). The DAG is represented in Figure 1, seeking to determine which variables should be adjusted for in the analysis, minimizing the risk of over-adjustment or omission of relevant confounders. In addition, the title of Figure 1 has been revised as suggested and now reads:

Revised text:

"Figure 1. Representation of the Directed Acyclic Diagram (DAG), created to identify potential confounders in the relationship between functional dentition and consumption of ultra-processed foods."

Reference: Tennant PWG, Murray EJ, Arnold KF, et al. Use of directed acyclic graphs (DAGs) to identify confounders in applied health research: review and recommendations. International Journal of Epidemiology. 2021;50(2):620-632. doi:10.1093/ije/dyaa213

In addition, we have added a "Conclusion" section at the end of the discussion.

Reviewer #2: The manuscript is technically sound and the data support the conclusions. However, more clarification about th

---

## [Decision Letter · Decision Letter 1]

Dear Dr. Hugo,

Thank you for submitting your manuscript to PLOS ONE. After careful consideration, we feel that it has merit but does not fully meet PLOS ONE’s publication criteria as it currently stands. Therefore, we invite you to submit a revised version of the manuscript that addresses the points raised during the review process.

We look forward to receiving your revised manuscript.

Kind regards,

Hadi Ghasemi

Academic Editor

PLOS ONE

Journal Requirements:

Reviewers' comments:

Reviewer's Responses to Questions

**Comments to the Author**

Reviewer #1: (No Response)

Reviewer #2: All comments have been addressed

Reviewer #3: (No Response)

2. Is the manuscript technically sound, and do the data support the conclusions?

Reviewer #1: Yes

Reviewer #2: Yes

Reviewer #3: Yes

3. Has the statistical analysis been performed appropriately and rigorously?

Reviewer #1: I Don't Know

Reviewer #2: Yes

Reviewer #3: Yes

4. Have the authors made all data underlying the findings in their manuscript fully available?

Reviewer #1: Yes

Reviewer #2: Yes

Reviewer #3: Yes

5. Is the manuscript presented in an intelligible fashion and written in standard English?

Reviewer #1: Yes

Reviewer #2: Yes

Reviewer #3: Yes

Reviewer #1: Thank you for addressing previous comments.

Just two small ones:

1. You have repeated the full term for PNS in abstract. Once is enough.

2. Please add "Brazilian" before "adults" in the title of Table 1, as each table should be self-standing.

Reviewer #2: (No Response)

Reviewer #3: I have reviewed the revised version of the manuscript along with the authors’ responses to the editorial and reviewer comments. I appreciate the considerable effort the authors have undertaken to improve the clarity, methodological rigor, and presentation of the manuscript. The clarifications added to the methods section, improved articulation of the hypotheses, and refinement of the discussion enhance the manuscript’s scientific value and readability.

However, a few important issues remain and should be addressed before the manuscript can be considered for publication:

1. Descriptive Statistical Testing – Table 1

While the authors have now included Table 1, the p-values for comparisons across demographic and socioeconomic groups remain absent. In the initial review, I requested clarification on whether differences observed in ultra-processed food consumption across subgroups (e.g., age, sex, education) were statistically significant. Please include p-values in Table 1 to indicate whether the observed differences are statistically significant. If not included in the table, a supplementary note summarizing the key significant comparisons would suffice. This information is crucial for interpreting group-level variation and supports the argument that social determinants play a central role.

2. Figure 1 Caption and Terminology

The title of Figure 1 still reads as a sentence ("Representation of the Directed Acyclic Diagram (DAG)...") rather than a concise, telegraphic label. Please revise the figure caption to reflect standard scientific format. A recommended revision might be:

"Figure 1. Directed acyclic graph (DAG) illustrating hypothesized confounders in the relationship between functional dentition and ultra-processed food consumption."

3. Additional Discussion of Socioeconomic Findings (Optional but Recommended)

Although the authors explain that covariates were included as confounders, it would strengthen the discussion to briefly summarize which socioeconomic factors (e.g., education, income) were most strongly associated with ultra-processed food consumption in the adjusted model—even if these were not the primary variables of interest. Consider briefly commenting on the adjusted prevalence ratios for key covariates to contextualize the social gradient in food consumption and reinforce the policy implications.

4. Minor Language Polishing

The authors have addressed many language concerns, and the manuscript reads more clearly. However, I recommend a final round of professional English editing to ensure fluency, especially in the Abstract and Discussion sections.

**Do you want your identity to be public for this peer review?** For information about this choice, including consent withdrawal, please see our Privacy Policy

Reviewer #1: No

Reviewer #2: **Yes: ** Assistant Prof. Dr. Hanan Fadhil Alautry

Reviewer #3: **Yes: ** Mahsa Malekmohammadi

---

## [Author Response · Author response to Decision Letter 2]

8 May 2025

Review Comments to the Author

Reviewer #1: Thank you for addressing previous comments.

Just two small ones:

1. You have repeated the full term for PNS in abstract. Once is enough.

Response: We appreciate your observation. We have revised the text as suggested, using the acronym PNS after its first full mention in the abstract.

2. Please add "Brazilian" before "adults" in the title of Table 1, as each table should be self-standing.

Response: Thanks for the feedback, the term has been added.

Reviewer #2: (No Response)

Reviewer #3: I have reviewed the revised version of the manuscript along with the authors’ responses to the editorial and reviewer comments. I appreciate the considerable effort the authors have undertaken to improve the clarity, methodological rigor, and presentation of the manuscript. The clarifications added to the methods section, improved articulation of the hypotheses, and refinement of the discussion enhance the manuscript’s scientific value and readability.

However, a few important issues remain and should be addressed before the manuscript can be considered for publication:

1. Descriptive Statistical Testing – Table 1

While the authors have now included Table 1, the p-values for comparisons across demographic and socioeconomic groups remain absent. In the initial review, I requested clarification on whether differences observed in ultra-processed food consumption across subgroups (e.g., age, sex, education) were statistically significant. Please include p-values in Table 1 to indicate whether the observed differences are statistically significant. If not included in the table, a supplementary note summarizing the key significant comparisons would suffice. This information is crucial for interpreting group-level variation and supports the argument that social determinants play a central role.

Response: We appreciate the suggestion to include statistical significance in Table 1. In response, we have added unadjusted prevalence ratios (PRs) for all demographic and socioeconomic comparisons in the table. All PRs with a p-value <0.05 have been highlighted in bold and explicitly mentioned in the table footnote. This adjustment ensures clarity in identifying which subgroup differences in ultra-processed food consumption are statistically significant, reinforcing the role of social determinants as discussed in the manuscript. We believe this addresses the concern while maintaining readability.

2. Figure 1 Caption and Terminology

The title of Figure 1 still reads as a sentence ("Representation of the Directed Acyclic Diagram (DAG)...") rather than a concise, telegraphic label. Please revise the figure caption to reflect standard scientific format. A recommended revision might be:

"Figure 1. Directed acyclic graph (DAG) illustrating hypothesized confounders in the relationship between functional dentition and ultra-processed food consumption."

Response: Thanks for the suggestion, the title has been changed as suggested to "Figure 1. Directed acyclic graph (DAG) illustrating hypothetical confounders in the relationship between functional dentition and ultra-processed food consumption."

3. Additional Discussion of Socioeconomic Findings (Optional but Recommended)

Although the authors explain that covariates were included as confounders, it would strengthen the discussion to briefly summarize which socioeconomic factors (e.g., education, income) were most strongly associated with ultra-processed food consumption in the adjusted model—even if these were not the primary variables of interest. Consider briefly commenting on the adjusted prevalence ratios for key covariates to contextualize the social gradient in food consumption and reinforce the policy implications.

Response: We appreciate the suggestion. As recommended, we expanded the discussion to include a brief description of the main findings related to socioeconomic variables. In the revised text, we highlight that lower levels of education and income were positively associated with the consumption of ultra-processed foods, even after adjusting for functional dentition and other covariates. In addition, we have included a new Appendix I, which presents the complete results of the adjusted analysis, including prevalence ratios and their respective confidence differences for all variables included in the model. This table allows a detailed visualization of the covariates incorporated in the multivariate regression.

Revised text:

3rd paragraph of the discussion

It is important to consider that socioeconomic and behavioral factors may play a more significant role in food choice than the number of teeth.20 The inclusion of covariates such as income, education, urban/rural setting, and use of dental services in the analysis may have diluted the association between functional dentition and diet, as these factors can act as direct determinants in the choice of ultra-processed foods.7,23 The results also revealed that lower levels of education and income were consistently associated with higher consumption of ultra-processed foods, even after adjusting for dental status and other covariates, reflecting a clear social gradient. This finding reinforces the notion that structural determinants of health strongly influence dietary behavior, independent of oral health status (Appendix I). Eating behavior, shaped by cultural norms and economic constraints, may mediate this relationship, suggesting that a more comprehensive exploration of these pathways would be valuable in future research.24 – 26

4. Minor Language Polishing

The authors have addressed many language concerns, and the manuscript reads more clearly. However, I recommend a final round of professional English editing to ensure fluency, especially in the Abstract and Discussion sections.

Response: We thank the reviewer for the careful reading. In response to the recommendation, the manuscript has undergone a final language revision by a professional English. We have carefully revised the text, with particular attention to the Abstract and Discussion sections, to improve clarity, coherence, and overall fluency.

---

## [Decision Letter · Decision Letter 2]

Association between functional dentition and ultra-processed food consumption in Brazilian adults: a cross-sectional study

PONE-D-24-56465R2

Dear Dr. Hugo,

We’re pleased to inform you that your manuscript has been judged scientifically suitable for publication and will be formally accepted for publication once it meets all outstanding technical requirements.

Kind regards,

Hadi Ghasemi

Academic Editor

PLOS ONE

Additional Editor Comments (optional):

Reviewers' comments:

Reviewer's Responses to Questions

**Comments to the Author**

Reviewer #1: All comments have been addressed

Reviewer #3: All comments have been addressed

2. Is the manuscript technically sound, and do the data support the conclusions?

Reviewer #1: (No Response)

Reviewer #3: Yes

3. Has the statistical analysis been performed appropriately and rigorously?

Reviewer #1: (No Response)

Reviewer #3: Yes

4. Have the authors made all data underlying the findings in their manuscript fully available?

Reviewer #1: (No Response)

Reviewer #3: Yes

5. Is the manuscript presented in an intelligible fashion and written in standard English?

Reviewer #1: (No Response)

Reviewer #3: Yes

Reviewer #1: (No Response)

Reviewer #3: Thank you for your efforts, this work is of value and thanks for your contribution. All my comments were addressed. Some small typos remain.

**Do you want your identity to be public for this peer review?** For information about this choice, including consent withdrawal, please see our Privacy Policy

Reviewer #1: No

Reviewer #3: **Yes: ** Mahsa Malekmohammadi

---

## [Editor Report · Acceptance letter]

PONE-D-24-56465R2

PLOS ONE

Dear Dr. Hugo,

I'm pleased to inform you that your manuscript has been deemed suitable for publication in PLOS ONE. Congratulations! Your manuscript is now being handed over to our production team.

Kind regards,

on behalf of

Dr. Hadi Ghasemi

Academic Editor

PLOS ONE